# Effects of the Emitted Light Spectrum of Liquid Crystal Displays on Light-Induced Retinal Photoreceptor Cell Damage

**DOI:** 10.3390/ijms20092318

**Published:** 2019-05-10

**Authors:** Chao-Wen Lin, Chung-May Yang, Chang-Hao Yang

**Affiliations:** Departments of Ophthalmology, National Taiwan University Hospital, No.7, Zhongshan South Road, Taipei 100, Taiwan; b91401108@ntu.edu.tw (C.-W.L.); chungmay100@gmail.com (C.-M.Y.)

**Keywords:** photo-injury, liquid crystal display (LCD), light-emitting diodes (LED), blue light, light spectrum, photoreceptor, oxidative stress, NF-κB

## Abstract

Liquid crystal displays (LCDs) are used as screens in consumer electronics and are indispensable in the modern era of computing. LCDs utilize light-emitting diodes (LEDs) as backlight modules and emit high levels of blue light, which may cause retinal photoreceptor cell damage. However, traditional blue light filters may decrease the luminance of light and reduce visual quality. We adjusted the emitted light spectrum of LED backlight modules in LCDs and reduced the energy emission but maintained the luminance. The 661W photoreceptor cell line was used as the model system. We established a formula of the ocular energy exposure index (OEEI), which could be used as the indicator of LCD energy emission. Cell viability decreased and apoptosis increased significantly after exposure to LCDs with higher emitted energy. Cell damage occurred through the induction of oxidative stress and mitochondrial dysfunction. The molecular mechanisms included activation of the NF-κB pathway and upregulation of the expression of proteins associated with inflammation and apoptosis. The effect was correlated with OEEI intensity. We demonstrated that LCD exposure-induced photoreceptor damage was correlated with LCD energy emission. LCDs with lower energy emission may, therefore, serve as suitable screens to prevent light-induced retinal damage and protect consumers’ eye health.

## 1. Introduction

Computers and consumer electronics play important roles in modern society; however, the majority of these products use a liquid crystal display (LCD) as the screen with light-emitting diode (LED)-based backlight modules being widely used as the LCD light source. LEDs emit higher levels of blue light with shorter wavelengths than conventional light sources and are accompanied by an enhanced risk of photo-injury. Extensive blue light exposure also disrupts circadian rhythms [1], which may constitute a health hazard for humans. For example, nighttime lighting with blue light and the desynchronization of circadian rhythms may increase the incidences of psychiatric disorders, diabetes, obesity, and several kinds of cancers [2]. 

The photoreceptors and retinal pigment epithelial (RPE) cells of the retina comprise the main sites of light energy absorption [3,4]. It has been demonstrated that lights with shorter wavelengths induce more retinal damage than those with longer wavelengths [5,6,7,8]. In particular, LED-induced retinal photoreceptor and RPE cell damage have been observed in several animal studies [9,10], even at domestic lighting levels [11]. Epidemiologic and experimental evidence has further revealed that short-wavelength light exposure may also contribute to the pathogenesis of age-related macular degeneration (AMD) [12,13,14]. Notably, previous studies found that the mechanism of retinal injury is related to the generation of reactive oxygen species (ROS) and inflammatory reactions, which induce oxidative stress and cell apoptosis within the retina [15,16,17,18,19].

Light-induced retinal injury has received considerable research attention owing to the increasing societal exposure to display terminals that emit extensive levels of blue light. Nevertheless, although advances have been made related to treatments for e.g., nonexudative and exudative AMD, their effectiveness remains limited. In order to prevent light-induced damage of the retina and the alteration of circadian rhythms, blue light filters, blocking lenses, and blue light–filtering intraocular lenses have been widely used [20,21,22]. However, traditional filters or blocking lenses may decrease the luminance of light and reduce the contrast sensitivity and visual quality while performing visual tasks. Alternatively, as the severity of light-induced retinal damage is related to exposure time and the emitted energy, adjustment of the phosphor ratio of LED backlight modules in LCDs and modification of the visible light spectrum might allow maintenance of the luminance of the light source while reducing the energy exposure of the eye. Although the concept of tuning the emitted wavelength of LEDs in a smartphone LCD backlight to reduce the unhealthy effect of blue light has been suggested [23], the hypothesized effect was not confirmed by experiments. Therefore, identifying a method to alleviate the blue-light hazards while simultaneously maintaining the visual quality of the screen constitutes an important issue in environmental health perspectives.

In this study, we established a formula of the ocular energy exposure index (OEEI), which equals the total radiance of the LCD light spectrum divided by total luminance of the LCD. This index was similar to the concept of light hazard ratio in a previous study [24] and can be used as an indicator of LCD energy emission. We utilized 661W photoreceptor cell line as the model system, as these have been previously exploited in several studies of light-induced retinal damage [25,26,27,28,29]. We also modulated the emitted light spectrum and designed the LCDs with low, medium, and high OEEI value. The aim of this study was to investigate the cell viability, mitochondrial damage, ROS production, and apoptosis of photoreceptor cells upon exposure to LCDs with different OEEI values and elucidate the detailed mechanisms by which the emitted light spectrum leads to light-induced photoreceptor cell damage.

## 2. Results

### 2.1. 661 W Cell Death upon Exposure to Differing-Luminance LCDs

In order to determine the appropriate conditions for light-induced retinal photoreceptor cell damage, we investigated the viability of 661W with different duration (1, 2, and 3 days) and luminance (100, 200, and 300 nits) of LCD exposure. Cell viability was analyzed using alamarBlue assays. Cell viability decreased significantly on day 3 after exposure to 300 nits LCD (Figure 1). No significant cell death was observed upon exposure to LCDs with lower luminance.

### 2.2. 661 W Cell Viability upon Exposure to Differing-OEEI LCDs

We investigated the viability of 661W cells exposed to LCDs with different OEEI (low, medium, high) for 3 days. The luminance of all LCDs was 300 nits. Cell viability was analyzed using alamarBlue assays. Compared to the low OEEI group, cell viability decreased significantly after exposure to LCDs with medium and high OEEI (Figure 2). LCDs with higher OEEI induced increased levels of 661W cell death. 

### 2.3. 661 Cell Apoptosis upon Exposure to Differing-OEEI LCDs

We analyzed the damage of 661w cells exposed to LCDs with different OEEI for 24 h using the Terminal deoxynucleotidyl transferase-mediated dUTP-biotinide end labeling (TUNEL) assay. Increased numbers of apoptotic cells appeared in the higher OEEI groups (Figure 3A). The number of TUNEL-positive cells increased markedly after exposure to LCDs with higher OEEI (Figure 3B).

### 2.4. 661 W Cell ROS Production upon Differing-OEEI LCD Exposure

The mechanism of light-induced retinal injury may be related to oxidative stress. In order to detect the oxidative stress in 661W cells exposed to LCDs with different OEEIs, a ROS assay was used to assess ROS production. LCD exposure led to ROS generation in 661W cells. ROS production increased considerably after exposure to LCDs with higher OEEIs (Figure 4). The level of ROS production was correlated with OEEI intensity.

### 2.5. 661 W Cell Mitochondrial Damage upon Differing-OEEI LCD Exposure

We analyzed the extent of mitochondrial dysfunction using JC-1 staining. JC-1 forms J-aggregates in healthy cells, but is maintained in monomeric form in apoptotic cells. LCD exposure induced a decrease in mitochondrial membrane potential in 661W cells. Exposure to LCDs with higher OEEIs markedly decreased JC-1 aggregation and increased JC-1 monomers in 661W cells (Figure 5). The effect was also OEEI-intensity dependent. 

### 2.6. LCD-exposed 661W Cell Oxidative Stress/Inflammatory Response-associated Protein Expression

Light-induced retinal damage may be associated with oxidative stress and inflammatory reaction. To elucidate the mechanism through which LCD induced cell damage, we examined the protein expression associated with oxidative stress and inflammatory response. Exposure to LCD for 3 days led to significantly increased protein expression of intercellular adhesion molecule 1 (ICAM-1), inducible nitric oxide synthase (iNOS), monocyte chemoattractant protein 1 (MCP-1), and heme oxygenase-1 (HO-1) (Figure 6). The expression levels were correlated with OEEI intensity.

### 2.7. LCD-exposed 661W Cell Apoptosis-related Protein Cleaved Caspase-3 Expression

As LCD exposure induced apoptosis in 661w cells in an OEEI intensity-dependent manner, we evaluated the expression of the apoptosis-related protein cleaved caspase-3 in 661W cells after exposure to LCD with low, medium, and high OEEI for 3 days. In medium and high OEEI groups, the cleavage forms of caspase-3 were obviously observed (Figure 7) whereas this form was not seen in the low OEEI group.

### 2.8. Nuclear Factor-κB (NF-κB) Pathway Activation in LCD-exposed 661W Cells

As exposure to LCD with higher OEEI upregulated the expression of proteins regulated by NF-κB, we further investigated the interaction between OEEI intensity and the NF-κB pathway. NF-κB binds to specific DNA sequences to activate downstream gene expression. Therefore, electrophoretic mobility shift assay (EMSA) was performed to study the DNA-binding activity of NF-κB in LCD-exposed 661W cells. Compared to that in the low OEEI group, DNA-binding activity was markedly increased and NF-κB pathway was activated in the medium and high OEEI groups (Figure 8).

## 3. Discussion

The use of consumer electronics incorporating LCD-based screens including computers, laptops and smartphones is continuing to increase. However, the LED backlight modules emit extensive blue light that can damage retinal photoreceptors and RPE cells and alter circadian rhythms owing to its short wavelength and high energy, eventually becoming a health hazard. Here, we investigated the effects of the emitted light spectrum of LCDs on LED-induced retinal photoreceptor cell damage and elucidated the detailed mechanisms. As traditional blue light filters and blocking lenses may decrease the luminance of light and reduce the color and contrast sensitivity and visual quality, we alternatively designed LCDs with differing energy emission but the same luminance by adjusting the phosphor ratio and modulating the emitted light spectrum. We also established an index of LCD energy emission, termed OEEI, the value of which is represented by the radiant flux produced by each luminous flux and could be used to evaluate the light hazards. The results of the present study indicated that LCDs with higher OEEI caused stronger light-induced photoreceptor cell damage through the production of ROS and activation of the NF-κB pathway, along with upregulation of protein expression associated with inflammatory response and apoptosis.

In particular, exposure to an LCD with luminance of 300 nits for three days decreased the viability of 661W photoreceptor cells. However, compared to the effects of the LCD with low OEEI, LCD with higher OEEI induced increased levels of cell death and the percentage of apoptotic cells was also higher. Moreover, the effect was OEEI-intensity dependent. Based on the analysis of the emitted light spectrum, the major difference among the three OEEI groups was the radiance of shorter wavelength blue light and green light. Blue light energy emission was much higher in the high OEEI group than that in the medium and low OEEI groups. Although green light energy emission was also higher in the high OEEI group, the photoreceptor-derived cell damage caused by blue light was much more serious than that caused by green light [29]. LCD exposure-induced photoreceptor cell damage could, therefore, be mainly attributed to the level of emitted blue light energy. 

Our study demonstrated that the decline of cell viability after LCD exposure was due to elevated cellular ROS formation and mitochondrial dysfunction. Oxygen radicals initiate free radical chain reactions and the oxidative stress results in cellular dysfunction and cell death [30,31]. The effect is stronger in the outer retina because choriocapillaries provide higher concentrations of oxygen. Therefore, 661W photoreceptor cells constitute a good model for studying these effects. In addition, mitochondrial dysfunction can serve as a predictor of cell injury and apoptosis as the mitochondrion is essential for cellular physiological functions and energy production. In the present study, JC-1 staining was used as the indicator of mitochondrial membrane potential damage [32]. Light induces retinal injury through the production of free radicals, oxidative stress, and the damage of mitochondrial membrane potential, as previously reported [15,16,33,34]. The results of our study revealed that lower OEEI exposure produced less ROS, induced less oxidative stress and mitochondrial damage, and subsequently led to decreased apoptosis and cell death compared to that from high OEEI exposure.

NF-κB consists of a family of transcription factors which play critical roles in immunity, inflammation, cell proliferation, differentiation, and cell survival. In order to elucidate the molecular mechanism of the increased oxidative stress and cell apoptosis caused by LCD with higher OEEI, we used western blot analysis and EMSA to evaluate the expression of inflammation and apoptosis-related proteins and the modulation of the NF-κB system. The results of our study revealed that the expression of proteins associated with oxidative stress, inflammation, and the apoptotic pathway all increased in 661W cells exposed to LCD with higher OEEI. In particular, iNOS and HO-1 comprise oxidative stress-related proteins that could be induced in an oxidative environment such as blue light exposure [35,36]. Both ICAM-1 and MCP-1 are pro-inflammatory cytokines that facilitate leukocyteendothelial transmigration, enhance inflammatory vascular permeability, advance cellular extravasation reaction, and promote signal transduction to produce inflammatory effects [37,38,39,40]. Therefore, ICAM-1 and MCP-1 function as strong pro-inflammatory mediators and could be used as indicators to assess the inflammatory status. In comparison, caspase-3 is a marker of cell apoptosis; notably, the cleavage forms of caspase-3 were obviously observed in the group with higher OEEI exposure. Among these, ICAM-1, iNOS, and, MCP-1 are regulated by NF-κB [41,42,43]; in turn, the NF-κB system can be induced by ROS production [44,45]. The result of EMSA in our study demonstrated that exposure to LCDs with higher OEEI activated the NF-κB pathway. Taken together, the findings of the present study suggested that LCD exposure induced oxidative stress and then upregulated the NF-κB pathway. Consequently, it further upregulated NF-κB downstream genes and enhanced the expression of oxidative stress, inflammation, and apoptosis-related proteins, finally inducing cell death. The effect was correlated with OEEI intensity, which also represented LCD energy emission, especially that of blue light. 

Our study had some limitations. First, this was an in vitro study. We used 661W cell line as our model system and the cells were still dividing and not fully differentiated. 661W cells have the potential to differentiate into neuronal cells with the treatment of staurosporine [46]. Caspases are also involved in some non-apoptotic processes including cell differentiation [47]. The mechanisms of apoptosis and caspase-mediated cell death may not be the same as that in the well-differentiated human retinal photoreceptors. However, 661W cells express cone photoreceptor features and respond to light stimulation [48]. This cell line has been widely used as the model of light-induced retinal damage in several studies [25,26,27,28,29]. Based on our results, the cleavage forms of caspase-3 were obviously observed in medium and high OEEI group but not in the control or low OEEI group. It implied that apoptosis rather than cell division or differentiation played the major role in the expression of caspases. Further research with primary retinal cell culture or animal experiments may be needed to confirm our results. The exact effect on the human retina and other aspects of light hazards require further investigation. Second, the majority of our experiments were performed under the condition of 3-day exposure with a luminance of 300 nits. Our findings are not necessarily generalizable to other conditions such as higher luminance or longer exposure times. The chronic cumulative effects of daily blue light exposure may cause more photochemical damage. However, the luminance of displays for laptops and other mobile devices is usually between 200 and 300 nits on average. Our results therefore still provided useful information regarding LCD exposure-induced retinal damage in daily life. Third, the visible light spectrum and emitted energy spectrum of the LCDs in our experiments were fixed. We could not analyze the effect of other specific patterns of light spectra and determine the most appropriate light source to maintain eye health. Further studies with specific light spectrum modulation should be considered.

## 4. Materials and Methods

### 4.1. 661 W Cell Culture

The 661W cell line was obtained from Dr. M. Al-Ubaidi (Univ. of Houston). 661W constitutes a mouse photoreceptor cell line immortalized by the expression of simian virus 40 T antigen, which exhibits biochemical features characteristic of cone photoreceptor cells [48]. The 661W cells were cultured in Dulbecco’s modified Eagle’s media (DMEM) containing 10% phosphate buffer solution (PBS), and 1% penicillin-streptomycin (100 U/mL penicillin and 100 μg/mL streptomycin) at 37 °C and 5% CO_2_. Cells were passaged by trypsinization every 3-4 days. Cells were used at the second to fifth passages.

### 4.2. LCD Light Exposure

The 661W cells were exposed to LCDs with luminance of 0,100, 200, and 300 nits for morphological observation and to determine the suitable luminance and duration for subsequent experiments. 661W cells were seeded on a 24-well plate at a density of 5 × 10^4^/well, which was then placed on a special framework over the LCD (Figure 9). To ensure a stable growth environment for 661W cells, the device was placed inside the CO_2_ incubator. Temperature was maintained between 36.5 and 37.5 °C. 

### 4.3. Cell Viability Assay

Cell viability was assessed using the alamarBlue assay (Thermo Fisher Scientific, Waltham, MA, USA), which comprises a resazurin-based solution that functions as a cell health indicator. 661W cells were seeded on 24-well plates at a density of 5 × 10^4^/well, then 500 μL of alamarBlue was added to each well. Following incubation at room temperature in the dark for 4 h, 100 μL of supernatant from each well was transferred to a new 96-well plate. The absorbance was measured at a wavelength of 570 and 600 nm using a microplate reader (Bio-Rad Laboratories Inc., Hercules, CA, USA). Cell viability was estimated by normalizing the 570 nm value to the 600 nm value. 

### 4.4. Preparation of LCDs with Different Energy Spectra (OEEI)

We established a formula of OEEI as the total radiance of the LCD visible spectrum divided by total luminance of the LCD, as Equation (1):
(1)Ocular energy exposure index (OEEI)=RadianceLuminance=WSr∗m2nits=WSr∗m2lmSr∗m2(Wlm)
where *W*/*Sr*·*m*^2^ is the unit of total radiance of the LCD and nits is the unit of total luminance of the LCD. The unit of OEEI is *W*/*lm*. A lower OEEI value is represented by the lower radiant flux (*W*) produced by each luminous flux (*lm*), whereas a higher OEEI value is represented by the higher radiant flux (*W*) produced by each luminous flux (*lm*).

In this study, we used 21-inch LCDs with three different OEEI values. The spectrum energy of LCDs is adjusted by the phosphor ratio of the LED backlight modules. The visible light spectra of low, medium, and high OEEI value LCDs are shown in Figure 10. The radiance of shorter wavelength blue light and green light was higher in the high OEEI group than that in the medium and low OEEI groups. The difference was especially larger in blue light. As shown in Table 1, the OEEI value of LCDs with low, medium, and high OEEI could be separately represented by 3.35 × 10^−3^, 3.53 × 10^−3^, and 3.75 × 10^−3^ (W/lm). The luminance was maintained at 300 nits.

### 4.5. TUNEL Assay

The TUNEL assay was performed using a FragEL^TM^ DNA fragmentation detection kit (Calbiochem, Darmstadt, Germany). Tissue sections were deparaffinized, rehydrated, and blocked using endogenous peroxidase with H_2_O_2_ for 30 min. Antigen retrieval was achieved by pressure-cooking in a 0.1 M citrate buffer at pH 6 for 10 min followed by cooling at room temperature prior to incubation with the enzyme. The TUNEL enzyme (1 h at 37 °C) and peroxidase converter (30 min at 37 °C) were applied to the 10-μm sections after incubation for 5 min in a permeabilizing solution of 0.1% Triton-X in 0.1% sodium citrate. The fluorescent signals were obtained by adding FITC-Avidin, which bound to the biotinylated-dU of the damaged DNA. After staining, image analysis was used to quantify the relative fluorescence intensity of the TUNEL-positive cells, with the number of TUNEL-stained nuclei quantified in four random slides per sample.

### 4.6. Measurement of Intracellular ROS

Intracellular ROS in 661W cells was measured using a ROS assay. This nonfluorescent probe accumulates within cells and transforms into 2′,7′-dichlorodihydrofluorescein (H2DCF) by deacetylation, and then reacts with ROS to form the fluorescent compound dichlorofluorescein (DCF). First, 661W cells were plated in Petri dishes. After exposure to the LCD for 3 days, 10 mM of 2′,7′-dichlorodihydrofluorescein diacetate (H2DCFDA, Sigma-Aldrich, St. Louis, MO, USA) was added to the medium for 30 min at 37 °C in the dark. After two rinses with PBS, cells were visualized by fluorescence microscopy at excitation/emission wavelengths of 548/573 nm. Quantitative analysis of the images was performed using image-processing software (Image-J; National Institutes of Health).

### 4.7. Mitochondrial Membrane Potential Detection

The mitochondrial membrane potential was measured using the JC-1 mitochondrial membrane potential assay kit (Cayman Chemical Company, Ann Arbor, MI, USA). The 661W cells cultured in 24-well plates were exposed to LCD with low, medium, and high OEEI. After 3 days of exposure, 50 μL of JC-1 staining buffer was added to 1 mL of culture medium. JC-1 forms J-aggregates in healthy cells and remains in its monomeric form in apoptotic or unhealthy cells. J-aggregates and JC-1 monomers can be detected using settings designed to detect Texas Red and FITC, respectively. The images were captured using a fluorescence microscope, which was used to detect healthy cells with mainly JC-1 J-aggregates (excitation/emission, 540/605 nm) and apoptotic or unhealthy cells with mainly JC-1 monomers (excitation/emission, 480/510 nm). The number of cells (red or green-stained cells) were counted in a blind manner using image-processing software (Image-J).

### 4.8. Western Blot

Proteins were extracted from 661W cells by incubation in a lysis buffer consisting of 0.5 M Tris-HCl (pH 7.4), 1.5 M NaCl, 2.5% deoxycholic acid, 10% NP-40, 10 mM ethylenediaminetetraacetic acid (EDTA), and protease inhibitors (Complete Mini; Roche Diagnostics Corp., Rotkreuz, Switzerland). For western blot analysis, the protein samples were mixed 1:1 with Laemmli sample buffer and boiled for 5 min. Samples (equivalent to 50 μg total protein) were resolved by 10% sodium dodecyl sulfate-polyacrylamide gel electrophoresis. Separated proteins were electrophoretically transferred to polyvinylidene fluoride membranes (Immobilon-P) in a buffer containing 200 mM NaCl, 200 mM Tris-base, and 10 mM MgCl_2_ (pH 9.5). Nonspecific binding was blocked by incubation of membranes with 5% milk in PBS containing 0.1% Tween-20 (PBST) for 1 h at room temperature. The blots were then incubated with the following primary antibodies diluted in 5% milk in PBST overnight at 4 °C: goat anti-rat ICAM-1 (1:1000 dilution; R&D Systems Inc., Minneapolis, MN, USA; catalog no. AF583), rabbit anti-rat MCP-1 (1:1000; Abcam Inc., Cambridge, UK; catalog no. ab25124), rabbit anti-rat iNOS (1:1000; Abcam Inc., Cambridge, UK; catalog no. ab3523), rabbit anti-rat HO-1 (1:1000; Abcam Inc., Cambridge, UK; catalog no. ab13243), rabbit anti-rat caspase-3 (1:1000; Abcam Inc., Cambridge, UK; catalog no. ab4051) and mouse anti-rat GAPDH (1:5000; Abcam Inc., Cambridge, UK; catalog no. ab8224). After washing with PBST, the membranes were incubated with horseradish peroxidase-conjugated secondary antibody for 1 h and visualized by chemiluminescence. (Pierce Biotechnology, Waltham, MA, USA). The relative expression of proteins was quantified by densitometry scanning of blots with Image J software.

### 4.9. Nuclear Protein Extraction and NF-κB EMSA

The treated 661W cells were trypsinized, resuspended, and homogenized in buffer A [10 mM HEPES (pH 7.9), 1.5 mM KCl, 10 mM MgCl_2_, 1.0 mM dithiothreitol (DTT), and 1.0 mM phenylmethylsulfonyl fluoride (PMSF)]. The tissues were then homogenized (Dounce; Bellco Glass Co., Vineland, NJ, USA), followed by centrifugation at 5000 g at 4 °C for 10 min. The crude nuclear pellet was suspended in 200 μL of buffer B [20 mM HEPES (pH 7.9), 25% glycerol, 1.5 mM MgCl_2_, 420 mM NaCl, 0.5 mM DTT, 0.2 mM EDTA, 0.5 mM PMSF, and 4 μM leupeptin]. The sample was incubated on ice for 30 min and centrifuged at 12,000 g at 4 °C for 30 min. The supernatant containing the nuclear proteins was collected. A bicinchoninic acid assay kit, with bovine serum albumin as the standard (Pierce Biotechnology, Waltham, MA, USA), was used to determine the protein concentration.

### 4.10. NF-κB EMSA

EMSA was performed with an NF-κB DNA-binding protein detection system (Pierce Biotechnology, Waltham, MA, USA) according to the manufacturer’s instructions. We incubated a 10 μg nuclear protein aliquot with a biotin-labeled NF-κB consensus oligonucleotide probe (5′-AGTTGAGGGGACTTTCCCAGGC-3′) for 30 min in binding buffer and then determined the specificity of the DNA/protein binding by adding a 100-fold molar excess of unlabeled NF-κB oligonucleotide for competitive binding 10 min before adding the biotin-labeled probe.

### 4.11. Statistical Analysis

The values are shown as the means ± SD or median and the interquartile range in non-parametric tests. Statistical analysis for multiple comparisons in each study was determined by performing a Kurskal-Wallis test followed by post hoc Dunn test. Two-way analysis of variance (ANOVA) followed by Dunnett’s multiple comparisons test was used to examine the influence of luminance and duration of exposure on the cell viability. A p value ≤0.05 was considered significant. Statistical analysis was performed using SPSS software (version 17.0, SPSS Inc., Chicago, IL, USA).

## 5. Conclusions

The present study demonstrated that LCD exposure caused retinal photoreceptor cell damage that was correlated with the level of emitted energy, especially that of blue light. When LCDs emitted more energy; i.e., high OEEI LCD, the cell viability decreased to a greater degree and the percentage of apoptotic cells increased significantly. High energy light exposure induced the production of ROS, destroyed the mitochondrial function, activated the NF-κB pathway, and enhanced the expression of oxidative stress, inflammation, and apoptosis-related proteins. As consumer electronics are indispensable in daily life in the modern era of computing, the quest for increased image quality may be accompanied by higher energy emission of the light source, in turn resulting in more severe damage to the retina and enhancement of other blue light hazards such as circadian rhythm desynchronization. This represents a major societal health concern toward which traditional blue light filters and blocking lenses may be poorly suited, as consumers may be unwilling to sacrifice visual quality. Alternatively, in our present study, we modulated the visible light spectrum of the light source and reduced the emitted energy of the LCD. Consequently, the light-induced retinal photoreceptor cell damage was alleviated. LCDs with lower energy emission may, therefore, serve as more suitable screens to prevent light hazards to the human eye.

## Figures and Tables

**Figure 1 ijms-20-02318-f001:**
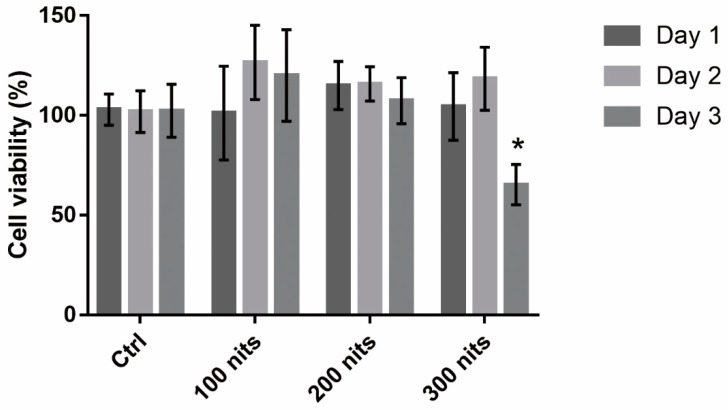
Viability of 661W cells exposed to liquid crystal displays with different luminance for 1, 2, and 3 days. Cell viability was analyzed using alamarBlue assays. Ctrl: control group, no light exposure; All data represent the means ± SD. * *p* < 0.05 compared to data on day 1 using two-way analysis of variance (ANOVA) with post hoc Dunnett’s multiple comparisons test; *n* = 16 per group.

**Figure 2 ijms-20-02318-f002:**
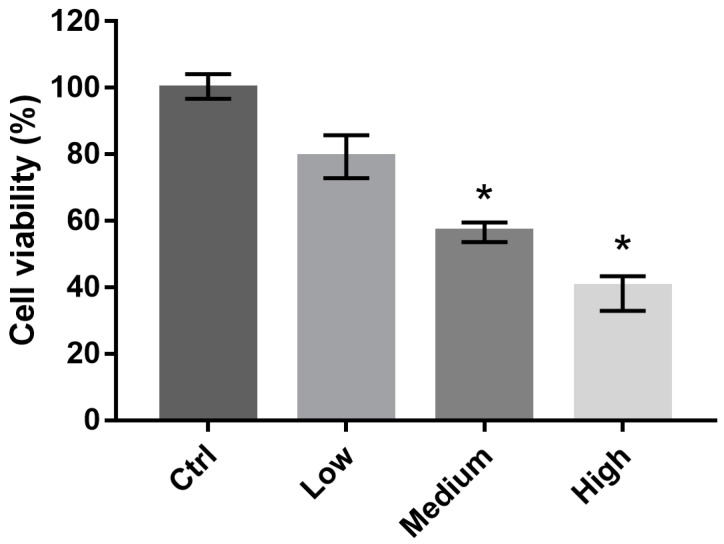
Viability of 661W cells exposed to liquid crystal displays with different ocular energy exposure index (OEEI) values for 3 days. Cell viability was analyzed using alamarBlue assays. Ctrl: control group, no light exposure; All data represent the median and the interquartile range. * *p* < 0.05 compared to the low OEEI group using the Kurskal-Wallis test with post hoc Dunn test; *n* = 12 per group.

**Figure 3 ijms-20-02318-f003:**
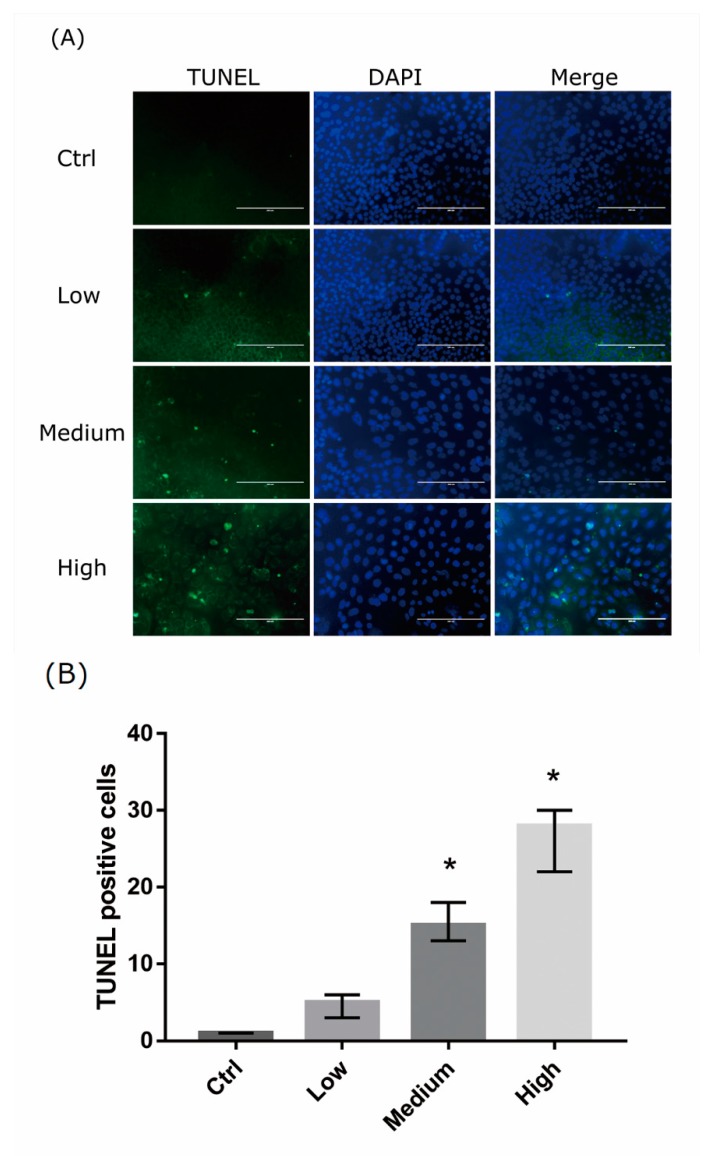
Apoptotic cells detected by terminal deoxynucleotidyl transferase-mediated dUTP-biotinide end labeling (TUNEL) in 661W cells. (**A**) Representative images of 661W cell apoptosis in controls and cells exposed to liquid crystal display (LCD) with low, medium, and high ocular energy exposure index for 24 h. Ctrl: control group, no light exposure; bar = 200 μm. (**B**) Number of TUNEL-positive cells was counted in at least four randomly chosen views, represented as columns. Ctrl: control group, no light exposure; All data represent the median and the interquartile range. * *p* < 0.05 compared to the control group using the Kurskal-Wallis test with post hoc Dunn test; *n* = 3 in each group.

**Figure 4 ijms-20-02318-f004:**
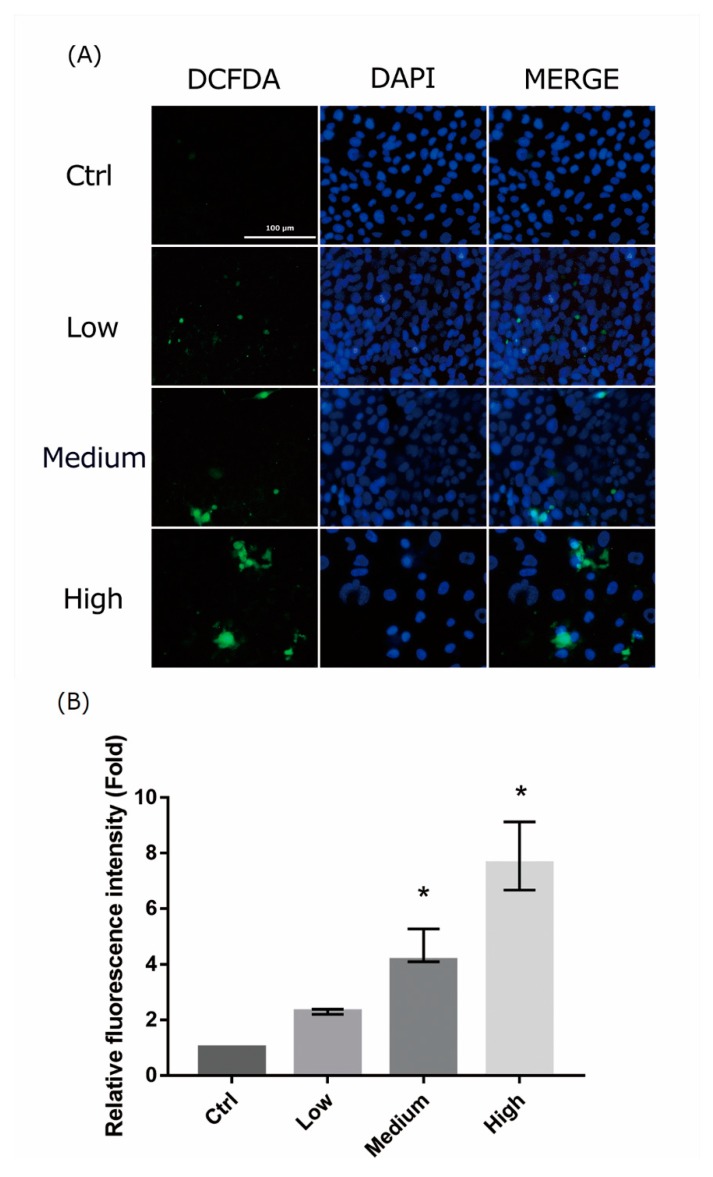
Reactive oxygen species assay in 661W cells after exposure to liquid crystal display (LCD) with different ocular energy exposure index (OEEI) for 3 days. (**A**) 2′,7′-dichlorodihydrofluorescein diacetate (H2DCFDA) was added to 661W cells after exposure to LCD with low, medium, and high OEEI. The images were captured by florescence microscopy. Bar = 100 μm. (**B**). Quantitative analysis of the images obtained from fluorescence microscopy using image-processing software (Image-J). Fluorescence level in the control group was arbitrarily set as 1. Ctrl: control group, no light exposure; All data represent the median and the interquartile range. * *p* < 0.05 compared to the control group by Kurskal-Wallis test with post hoc Dunn test; *n* = 3 in each group.

**Figure 5 ijms-20-02318-f005:**
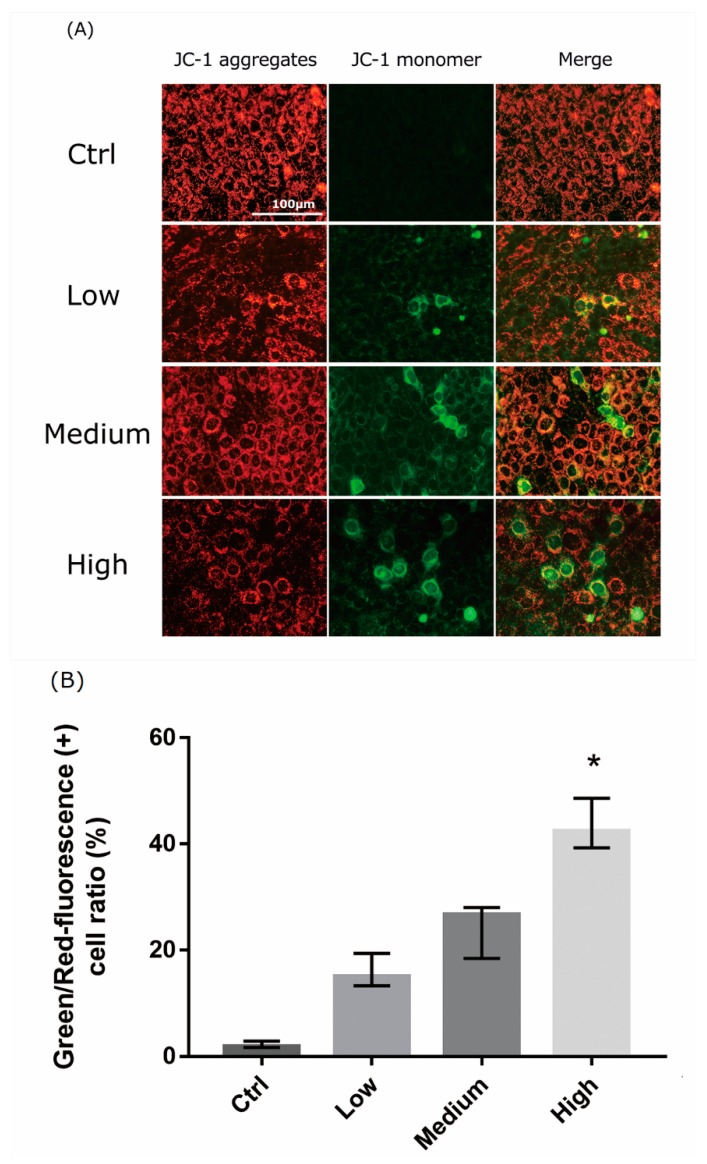
Liquid crystal display (LCD) exposure induces mitochondrial dysfunction in 661W cells. (**A**) 661W cells were subjected to JC-1 staining, which was analyzed using florescence microscopy. J-aggregates and JC-1 monomers were detected with settings designed to detect Texas Red and FITC, respectively. The 3-day exposure to LCD decreased JC-1 aggregation and increased JC-1 monomers in 661W cells. The effect was stronger in LCDs with higher ocular energy exposure index (OEEI). Bar = 100 μm. (**B**) Green/Red-fluorescence (+) cell ratio (%). The numbers of cells (red or green-stained cells) were counted using image processing software (Image-J). Ctrl: control group, no light exposure; All data represent the median and the interquartile range. * *p* < 0.05 compared to the control group by Kurskal-Wallis test with post hoc Dunn test; *n* = 4 in each group.

**Figure 6 ijms-20-02318-f006:**
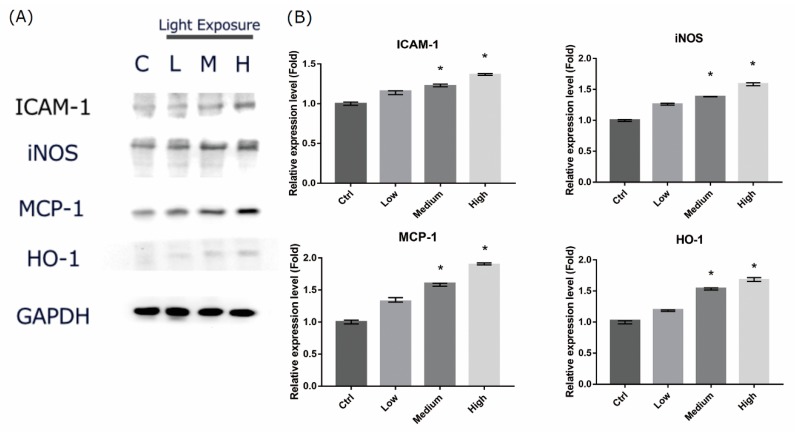
Liquid crystal display (LCD) exposure increases oxidative stress and inflammatory-related protein expression levels in 661W cells. (**A**) Evaluation of the protein expression of enzymes associated with inflammatory response by western blot analysis. 661W cells were exposed to LCD with low, medium, and high ocular energy exposure index (OEEI) for 3 days. GAPDH was used as the internal control. C: control group, no light exposure; L: low, M: medium, H: high OEEI. (**B**) Relative expression of intercellular adhesion molecule 1 (ICAM-1), inducible nitric oxide synthase (iNOS), monocyte chemoattractant protein 1 (MCP-1), and heme oxygenase-1 (HO-1). Ctrl: control group, no light exposure; All data represent the median and the interquartile range. * *p* < 0.05 compared to the control group by Kurskal-Wallis test with post hoc Dunn test; *n* = 5 per group.

**Figure 7 ijms-20-02318-f007:**
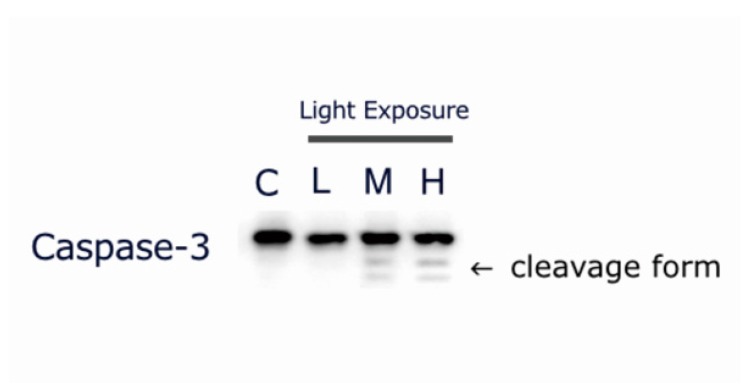
Exposure to liquid crystal display (LCD) with medium and high ocular energy exposure index (OEEI) increases cleaved caspase-3 expression in 661W cells. 661W cells were exposed to LCD with low, medium, and high OEEI for 3 days. The caspase-3 expression levels were analyzed by western blot. In medium and high OEEI groups, the cleavage forms of caspse-3 were obviously observed (arrow).

**Figure 8 ijms-20-02318-f008:**
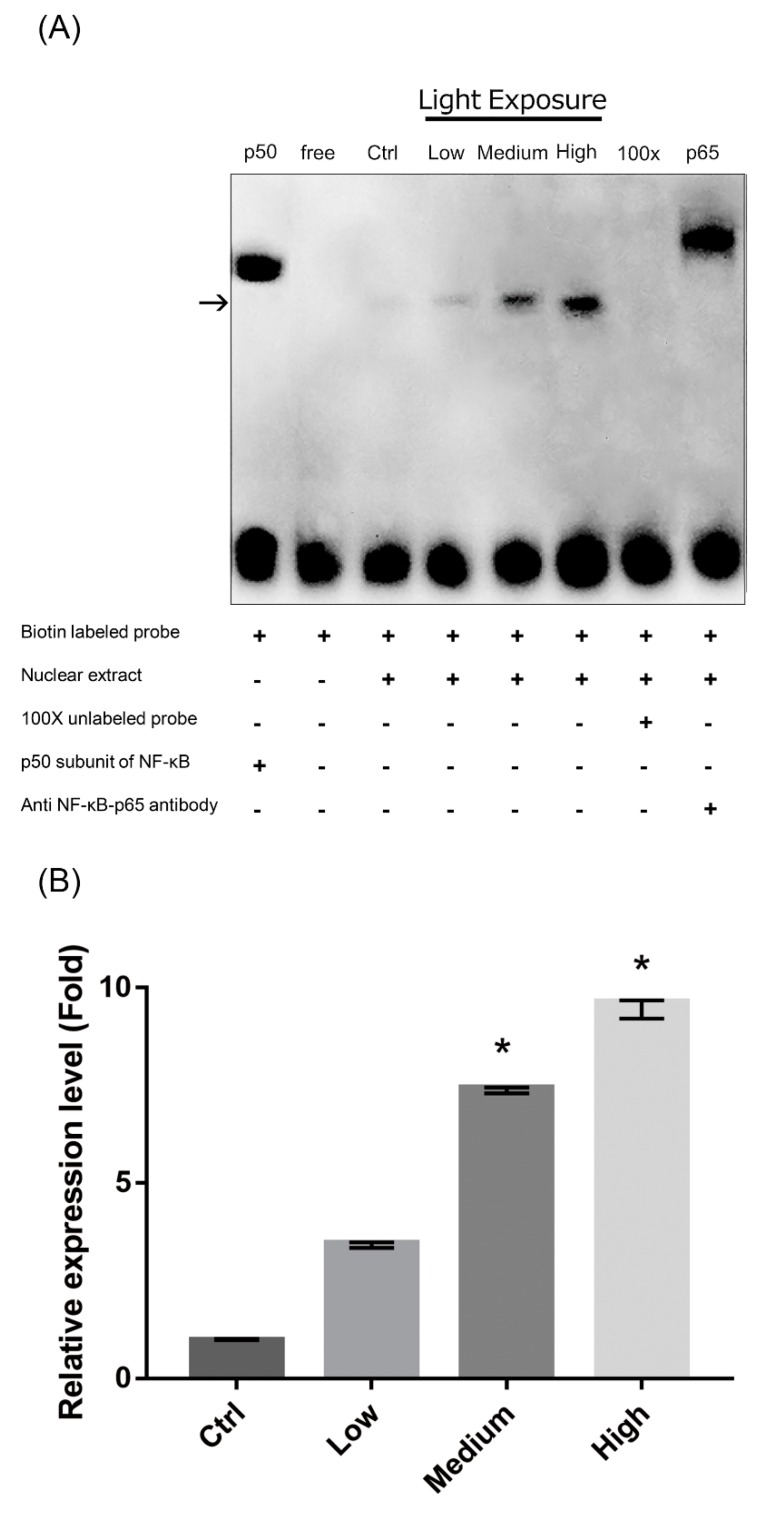
Exposure to liquid crystal display (LCD) with higher ocular energy exposure index (OEEI) activates the nuclear factor-κB (NF-κB) pathway. (**A**) Electrophoretic mobility shift assay (EMSA) was used to evaluate the DNA-binding activity of NF-κB in 661W cells after exposure to LCD with low, medium, and high OEEI for 3 days. Adding a 100-fold molar excess of unlabeled NF-κB probe completely inhibited the binding of labeled probe to the NF-κB /DNA complex. Lane 1: p50 subunit of NF-κB; Lane 2: free probe; Lane 3: control group, no light exposure; Lane 4: Low OEEI group; Lane 5: medium OEEI group; Lane 6: high OEEI group; Lane 7: competition with 100x unlabeled NF-κB probe; Lane 8: anti-p65 antibody supershift band. (**B**) Quantification of EMSA results. Expression level in the control group was arbitrarily set as 1. Ctrl: control group, no light exposure; All data represent the median and the interquartile range. * *p* < 0.05 compared to the low OEEI group using Kurskal-Wallis test with post hoc Dunn test; *n* = 3 in each group.

**Figure 9 ijms-20-02318-f009:**
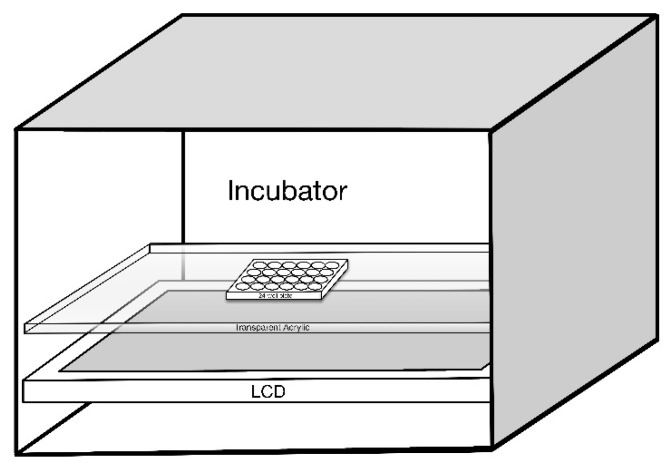
Schematic diagram of 661W cells seeded on a 24-well plate with the exposure of liquid crystal displays (LCD).

**Figure 10 ijms-20-02318-f010:**
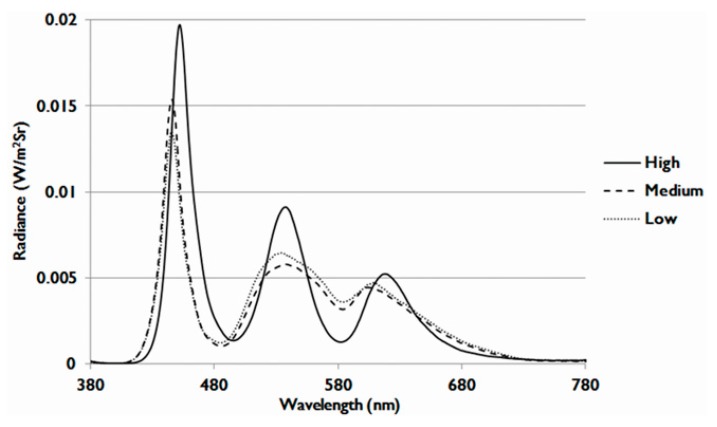
Visible light spectra of liquid crystal displays (LCDs) with low, medium, and high ocular energy exposure index (OEEI) values.

**Table 1 ijms-20-02318-t001:** Liquid crystal displays (LCDs) with low, medium, and high ocular energy exposure index (OEEI) values used in this study.

OEEI ^1^	Low	Medium	High
**Value**	3.35 × 10^−3^ (W/lm)	3.53 × 10^−3^ (W/lm)	3.75 × 10^−3^ (W/lm)

^1^ ocular energy exposure index.

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
