# Peer review of "Effects of the Emitted Light Spectrum of Liquid Crystal Displays on Light-Induced Retinal Photoreceptor Cell Damage"

_ijms, 2019, doi:10.3390/ijms20092318_

Round 1

Reviewer 1 Report

Several experimental evidences have indicated LED light induces retinal damages which might contribute to the pathogenesis of ocular disease. Authors used mouse derived photoreceptor cells (661W) as a model to investigate whether the exposure of LCD light in different emitting energy had any effects in the photoreceptor cell damage. Authors developed the ocular energy exposure index (OEEI) and used it as the indicator of LCD light exposure. They exposed LCD light to the cultured 661W for three days. They found the same luminance (300 nits) but higher LCD emitting energy decreased 661W cell viability. Their further investigation also indicates that the high energy light exposure induced higher ROS production, mitochondria dysfunction, and more expressions of inflammatory and apoptotic markers. They concluded high LCD light can be hazards to human eyes and lower energy emission of LCD may prevent retinal damages.

The study was original and all experiments were logically designed and well conducted. The results support the most of author’s conclusion. The manuscript is well written, but I think adding more details in results and methods would help for readers to follow the story easier. I also would like to make several comments and small suggestions.

1.       Interpretation of the results from the Fig.1 is not very clear. Were Fig. 1 experiments to find right exposure duration and luminance? Does day3 of 300 nits imply the medium intensity of Fig. 2 ?

2.       Authors should include more details about cultural condition of the 661W (i.e. how much cell were seeded and how many days cells were cultured before the LCD light exposure). Cell density may be critical to the experimental results.

3.       I think it is more appropriate to use One-way ANOVA (Maybe Two-way ANOVA for fig1.) for statistical analysis for fig 1.-3.

4.       For Fig. 8, inserting a table that shows what was added or not added (i.e. + or -) between the image and graphs would be beneficial. I am still not quite positive about how authors verify the bands were NF-kB binding activity. Do you think the mutation of binding site or non-saturated p65 antibody would be a good verification of NF-kB binding activity?

5.       The quality of image of Fig. 8 (A) is poor.

6.       In discussion, line 215, I think green wave was also higher in the high OEEI. Authors may include the study of Kuse et al., 2014 in their discussion so only blue light damages 661W cells.

7.       Photo of 661W cells were being exposed by the LCD light or a schematic diagram of how experiments were conducted would be very helpful for readers.

Author Response

Point 1: Interpretation of the results from the Fig.1 is not very clear. Were Fig. 1 experiments to find right exposure duration and luminance? Does day3 of 300 nits imply the medium intensity of Fig. 2 ?

Response 1: We appreciate the reviewer’s question. The aim of experiments showed in Fig.1 was to find out the appropriate exposure duration and luminance for further experiments. We have modified the sentence in line 75-76. In the experiments showed in Fig.2, we used LCDs with three different light spectra (low, medium, and high OEEI values). The luminance was maintained at 300 nits. This information was provided in Materials and Methods section. We appreciate the reviewer’s comment and added the description about the luminance of these LCDs in line 88.

Point 2: Authors should include more details about cultural condition of the 661W (i.e. how much cell were seeded and how many days cells were cultured before the LCD light exposure). Cell density may be critical to the experimental results.

Response 2: We appreciate the reviewer’s question. We agree that cell density is critical to the experimental results. Cells were passaged by trypsinization every 3-4 days. Cells were used at the second to fifth passages. 661W cells were seeded on a 24-well plate at a density of 5×104/well. We added the description in line 288-289, 293, 303.

Point 3: I think it is more appropriate to use One-way ANOVA (Maybe Two-way ANOVA for fig1.) for statistical analysis for fig 1.-3.

Response 3: We appreciate the reviewer’s advice. In figure 1, we examined the influence of luminance and duration of exposure on the 661W cell viability. We used two-way analysis of variance (ANOVA) followed by Dunnett's multiple comparisons test. In figure 2 and 3, we performed a Kurskal-Wallis test followed by post hoc Dunn test. We have modified the description in statistical analysis and figure legend in each figure. We are really grateful for the reviewer’s advice.

Point 4: For Fig. 8, inserting a table that shows what was added or not added (i.e. + or -) between the image and graphs would be beneficial. I am still not quite positive about how authors verify the bands were NF-kB binding activity. Do you think the mutation of binding site or non-saturated p65 antibody would be a good verification of NF-kB binding activity?

Response 4: We appreciate the reviewer’s advice and question. We are really grateful for the reviewer’s advice and inserted a table to demonstrate what were added in each lane. EMSA was performed with a commercialized NF-κB DNA-binding protein detection system. Biotin-labeled NF-κB consensus oligonucleotide probe was used to verify the DNA-binding activity of NF-κB. The free probe lane contained a single band corresponding to the unbound DNA fragment. The lane with a nuclear protein extract contained another band that represents the larger, less mobile complex of probe bound to protein which was shifted up on the gel since it moved more slowly. 100-fold molar excess of unlabeled NF-κB oligonucleotide was used to determine the specificity of the DNA/protein binding. For supershift assay, an anti NF-κB-p65 antibody that recognizes the protein could be added to the mixture to create an even larger complex with a greater shift and to identify the protein present in the protein–DNA complex. This is a well-established method to determine the NF-κB/DNA-binding activity and the activation of NF-κB pathway.

Point 5: The quality of image of Fig. 8 (A) is poor.

Response 5: We appreciate the reviewer’s comment. We adjust the contrast of Fig.8(A) and erase some noises without change the expression of the target band. The image quality of EMSA may not be as good as the result of western blotting. We tried our best and the target band could be properly quantificated.

Point 6: In discussion, line 215, I think green wave was also higher in the high OEEI. Authors may include the study of Kuse et al., 2014 in their discussion so only blue light damages 661W cells.

Response 6: We appreciate the reviewer’s comment. We agree that the radiance of blue light and green light was both higher in the high OEEI group than that in the medium and low OEEI groups. Blue light damaged the photoreceptor-derived cells more severely than green light. We have modified the description in line 217-221 and 320-322. We also included the excellent study done by Kuse Y et al. in 2014 in our reference. We are really grateful for the reviewer’s advice.

Point 7: Photo of 661W cells were being exposed by the LCD light or a schematic diagram of how experiments were conducted would be very helpful for readers.

Response 7: We appreciate the reviewer’s comment. We added the schematic diagram of 661W cells seeded on a 24-well plate with the exposure of liquid crystal displays (LCD) (Fig 9). We are really grateful for the reviewer’s advice and hope that the readers could understand how the experiments were conducted.

Reviewer 2 Report

This reviewer has two major concerns regarding this paper.

The first concerns the statistics methods used. Regarding the size of the samples non parametric test should be used and the results must be shown as the median and the interquartile range. No student test is valid because autors always compare more than two groups. Without proper statistics test I can not say which is the significance of the content of this paper. .

The second concern is related to the model used and the description of the mechanisms of cell death. The cells used are in division and therefore, not differentiatied, so that caspases are expressed wich is not the case in differenciated retina. This information needs to be dicussed in the context.

Author Response

Point 1: The first concerns the statistics methods used. Regarding the size of the samples non parametric test should be used and the results must be shown as the median and the interquartile range. No student test is valid because autors always compare more than two groups. Without proper statistics test I can not say which is the significance of the content of this paper.

Response 1: We appreciate the reviewer’s comment. In figure 1, we examined the influence of luminance and duration of exposure on the 661W cell viability. N=16 in each group and the data passed D'Agostino & Pearson normality test. Therefore, we used two-way analysis of variance (ANOVA) followed by Dunnett's multiple comparisons test. The values are shown as the means ± SD. In other results, regarding the size of sample, we performed a Kurskal-Wallis test followed by post hoc Dunn test. All data are shown as the median and the interquartile range in non-parametric tests. We have modified the description in statistical analysis and figure legend in each figure. We are really grateful for the reviewer’s advice.

Point 2: The second concern is related to the model used and the description of the mechanisms of cell death. The cells used are in division and therefore, not differentiatied, so that caspases are expressed wich is not the case in differenciated retina. This information needs to be dicussed in the context.

Response 2: We appreciate the reviewer’s comment. We admit that this 661W cell model system could not completely represent the effect on the human retina. It’s the limitation of in vitro study. The cells did not have the architecture of retina tissue and lack of functional influence of other retinal cell types. Besides, the cells were still in division and not fully differentiated. 661W cells have the potential to differentiate into neuronal cells with the treatment of staurosporine (please see reference 1). Caspases are also involved in some non-apoptotic processes including cell differentiation (please see reference 2). The mechanisms of apoptosis and caspase-mediated cell death may not be the same as that in the well-differentiated human retinal photoreceptors. However, 661W cells express cone photoreceptor features and respond to light stimulation (please see reference 3). This cell line has been widely used as the model of light-induced retinal damage in several studies (please see reference 4-8). Caspase-mediated cell death was also investigated in these researches. Besides, these cells did not receive any specific treatment except for LCD exposure, and we did not observe any specific morphologic change associated with cell differentiation. Based on our results, the cleavage forms of caspase-3 were obviously observed in medium and high OEEI group but not in control or low OEEI group. It implied that apoptosis rather than cell division or differentiation played the major role in the expression of caspases.

We are really grateful for the reviewer’s advice and have modified the description in the limitation part of “discussion” section (line 258-270).

Reference:

1.       Thompson, A.F.; Crowe, M.E.; Lieven, C.J.; Levin, L.A. Induction of Neuronal Morphology in the 661W Cone Photoreceptor Cell Line with Staurosporine. PLoS One 2015, 10, e0145270.

2.       Bell, R.AV.; Megeney, L.A. Evolution of caspase-mediated cell death and differentiation: twins separated at birth. Cell Death Differ 2017, 24(8):1359-1368.

3.       Tan, E.; Ding, X.Q.; Saadi, A.; Agarwal, N.; Naash, M.I.; Al-Ubaidi, M.R. Expression of cone-photoreceptor-specific antigens in a cell line derived from retinal tumors in transgenic mice. Invest Ophthalmol Vis Sci 2004, 45, 764–768.

4.       Santos, A.M.; Lopez-Sanchez, N.; Martin-Oliva, D.; de la Villa, P.; Cuadros, M.A.; Frade, J.M. Sortilin participates in light-dependent photoreceptor degeneration in vivo. PLoS one 2012, 7, e36243.

5.       Yang, L.P.; Zhu, X.A.; Tso, M.O. Role of NF-kappaB and MAPKs in light-induced photoreceptor apoptosis. Invest Ophthalmol Vis Sci 2007, 48, 4766–4776.

6.       Chen, W.J.; Wu, C.; Xu, Z.; Kuse, Y.; Hara, H.; Duh, E.J. Nrf2 protects photoreceptor cells from photo-oxidative stress induced by blue light. Exp Eye Res 2017, 154, 151-158.

7.       Ooe, E.; Kuse, Y.; Yako, T.; Sogon, T.; Nakamura, S.; Hara, H.; Shimazawa, M. Bilberry extract and anthocyanins suppress unfolded protein response induced by exposure to blue LED light of cells in photoreceptor cell line. Mol Vis 2018, 24, 621-632.

8.       Kuse, Y.; Ogawa, K.; Tsuruma, K.; Shimazawa, M.; Hara, H. Damage of photoreceptor-derived cells in culture induced by light emitting diode-derived blue light. Sci Rep 2014, 4, 5223.

Round 2

Reviewer 2 Report

The authors made the requested changes